

# Comparative transcriptome analysis of emerging young and mature leaves of *Bienertia sinuspersici,* a single-cell C4 plant

Richard M. Sharpe[1], Seanna Hewitt[2], Gerald Edwards[3] and Amit Dhingra[2]

[1] Department of Horticulture, Washington State University, Pullman, WA, United States of America
[2] Department of Horticultural Sciences, Texas A&M University, College Station, TX, United States of America
[3] School of Biological Sciences, Washington State University, Pullman, WA, United States of America

Corresponding author
Amit Dhingra, adhingra@tamu.edu

## ABSTRACT

**Background**. Efficient carbon capture by plants is crucial to meet the increasing demands for food, fiber, feed, and fuel worldwide. One potential strategy to improve the photosynthetic performance of plants is the conversion of $C_3$-type crops to $C_4$-type crops, enabling them to perform photosynthesis at higher temperatures and with less water. $C_4$-type crops, such as corn, possess a distinct Kranz anatomy, where photosynthesis occurs in two distinct cell types. Remarkably, *Bienertia sinuspersici* is one of the four known land plant species that perform $C_4$ photosynthesis within a single cell. This unique single-cell $C_4$ ($SCC_4$) anatomy is characterized by dimorphic chloroplasts and corresponding intracellular biochemistry. Because young, emergent *Bienertia* leaves first exhibit $C_3$ anatomy and then differentiate into the $C_4$ anatomy as the leaves mature, *Bienertia* represents an excellent system to explore the basis for a $C_3$ to $C_4$ transition.

**Methods**. To gain insight into the genes and pathways associated with the $C_3$ to $C_4$ transition between emerging young and mature *Bienertia sinuspersici* leaves, a comparative transcriptome analysis was conducted in which global gene expression and gene ontologies were compared between the two stages.

**Results**. In the emergent leaf, differentially expressed genes and enrichment of ontologies associated with the cell cycle and cytoskeletal dynamics were observed, while the mature leaf displayed enrichment of processes associated with photosynthesis and cellular energetics. Additionally, numerous transcription factors (TFs) associated with metabolic homeostasis, hormone and stress signaling, and developmental regulation were expressed throughout development, with unique TF expression profiles at each stage. These data expand our insights into the molecular basis of Binertia's unique cellular compartmentalization, chloroplast dimorphism, and single-cell C4 biochemistry and provide information that will be useful in the ongoing efforts to transform $C_3$-type crops into $C_4$ type.

## INTRODUCTION

The process of photosynthesis enables the conversion of light energy into chemical energy. Understanding photosynthetic biochemistry, structure, and function has enabled advancements in agriculture, including improved crop yields and resilience (*Sharma et al., 2016a*; *Sharma et al., 2016b*; *Pyc et al., 2017*; *Oleszek, Kowalska & Oleszek, 2019*; *Veiga et al., 2020*; *Van Vliet, Kronberg & Provenza, 2020*; *Liu et al., 2021*). The predominant form of photosynthesis, $C_3$ photosynthesis, produces a three-carbon organic molecule as the first product of carbon fixation *via* the ubiquitous enzyme ribulose-1,5-bisphosphate carboxylase/oxygenase (Rubisco) (*Ehleringer & Cerling, 2002*). However, Rubisco's catalytic inefficiency, particularly at higher temperatures, limits photosynthetic capacity as it favors binding oxygen over carbon dioxide, leading to photorespiration (*Yamori, Hikosaka & Way, 2014*; *Bräutigam & Gowik, 2016*). The occurrence of photorespiration makes $C_3$ photosynthesis a less efficient process under stress conditions. To overcome this limitation, plants have evolved alternative photosynthetic strategies, such as Crassulacean acid metabolism (CAM) and $C_4$ photosynthesis, which concentrate $CO_2$ around Rubisco and minimize $O_2$ competition. CAM achieves this by capturing $CO_2$ at night (*Yang et al., 2015*), while $C_4$ photosynthesis is characterized by the compartmentalization of carbon fixation across specialized cell types (*Ehleringer & Cerling, 2002*; *Edwards & Ogburn, 2012*; *Edwards, 2019*).

In canonical $C_4$ plants, such as maize, Kranz anatomy facilitates the spatial separation of photosynthetic processes between mesophyll and bundle sheath cells. In mesophyll cells, phosphoenolpyruvate carboxylase (PEPC) catalyzes the initial reaction between phosphoenolpyruvate (PEP) and bicarbonate ($HCO_3^-$), producing a four-carbon acid. This acid is transported to bundle sheath cells, where it is decarboxylated to release $CO_2$. The released $CO_2$ is then fixed by Rubisco *via* the $C_3$ (Calvin) cycle, which operates within the bundle sheath chloroplasts. This spatial organization minimizes photorespiration by maintaining a high concentration of $CO_2$ around Rubisco in the bundle sheath (*Muhaidat, Sage & Dengler, 2007*; *Bräutigam & Gowik, 2016*). Depending on the species, $C_4$ plants utilize one of three biochemical subtypes—NADP-malic enzyme (NADP-ME), NAD-malic enzyme (NAD-ME), or phosphoenolpyruvate carboxykinase (PEP-CK)—which differ in the decarboxylase enzyme employed (*Bräutigam et al., 2014*). The resulting three-carbon compound, pyruvate, is transported back to mesophyll cells, where pyruvate phosphate dikinase (PPDK) regenerates PEP, completing the cycle (*Kellogg, 2013*). The supporting enzyme, carbonic anhydrase (CA), catalyzes the conversion of $CO_2$ to bicarbonate, which serves as the substrate for PEPC (*Ehleringer & Cerling, 2002*). Together, these enzymes underpin the $CO_2$-concentrating mechanism characteristic of $C_4$ photosynthesis, enhancing photosynthetic efficiency under conditions of heat, drought, and low atmospheric $CO_2$.

While $C_4$ photosynthesis is commonly associated with Kranz anatomy, some plants can perform $C_4$ photosynthesis without it, including species of *Bienertia* and *Suaeda* (Chenopodiaceae) and certain aquatic plants, such as species of *Hydrilla* and *Egeria* (*Voznesenskaya et al., 2001*; *Voznesenskaya et al., 2002*; *Voznesenskaya et al., 2003*; *Bowes*

*et al., 2002*; *Freitag & Stichler, 2002*; *Akhani et al., 2005*; *Edwards & Voznesenskaya, 2011*; *von Caemmerer et al., 2014*). These non-Kranz $C_4$ plants employ alternative mechanisms to achieve spatial separation of carbon fixation. Among these, $SCC_4$ photosynthesis, which is observed in some members of the genera *Bienertia* and *Sueda* (*e.g.*, *Bienertia sinuspersici*, *Bienertia cycloptera*, *Bienertia kavirense*, and *Suaeda aralocaspica*), represents a unique adaptation where compartmentalization occurs within individual cells. In $SCC_4$, the division of photosynthetic roles is achieved by functionally distinct, dimorphic chloroplasts within the same cell. One group of chloroplasts catalyzes the initial fixation of bicarbonate into a four-carbon organic acid, which is then transported to a second group of chloroplasts housing NAD-ME and Rubisco. These processes are supported by the specialized localization of enzymes, the establishment of intracellular compartments, and cytoplasmic transport mechanisms. A side-by-side comparison of $C_3$, Kranz $C_4$, and $SCC_4$ photosynthetic pathways, as well as their compartmentalization, is presented in a previous publication (*Sharpe & Offermann, 2014*).

While both *Bienertia* and *Suaeda* are NAD-ME $C_4$ photosynthetic subtypes, as evidenced by $\delta^{13}C$, titratable acid, and $CO_2$ compensation assays (*Sharpe & Offermann, 2014*), they display distinct $SCC_4$ morphologies in regards to the way they employ dimorphic chloroplasts to achieve spatial separation of the $C_4$ pathway (*Voznesenskaya et al., 2003*; *Voznesenskaya et al., 2005*; *Edwards & Voznesenskaya, 2011*; *Langdale, 2011*; *Sharpe & Offermann, 2014*). In *Suaeda aralocaspica*, for example, dimorphic chloroplasts are localized to the distal and proximal poles of the cell in relation to the vascular tissue (*Koteyeva et al., 2016*). In contrast, in *Bienertia* species, dimorphic chloroplasts are located in a densely packed cytoplasmic compartment localized in the center of the cell, as well as in the cytoplasmic layer lying adjacent to the plasma membrane; these two chloroplast types, differing in thylakoid stacking and electron flow systems, are compartmentalized from each other by a large vacuole and are connected to each other *via* cytoplasmic strands traversing the vacuole (*Freitag & Stichler, 2000*; *Freitag & Stichler, 2002*; *Voznesenskaya et al., 2002*; *Mai et al., 2019*). It has been suggested that the shift of chloroplasts to the central cytoplasmic compartment may occur in response to light conditions (*Lara et al., 2006*; *Lara et al., 2008*).

Recent research in both *Suaeda* and *Bienertia* has provided insight into $SCC_4$ mechanisms and their potential for translational applications in agriculture. Studies in *Suaeda aralocaspica* have highlighted regulatory elements such as bHLH transcription factors that contribute to stress responses and photosynthetic adaptation, while isoform-specific investigations of PEPC have identified SaPEPC1 as critical for carbon fixation and abiotic stress tolerance (*Cao et al., 2021*; *Wei, Cao & Lan, 2022*). Additionally, *Suaeda*-derived *PEPC* genes have shown the potential to enhance drought tolerance and photosynthetic efficiency in $C_3$ crops (*Li et al., 2024a*; *Li et al., 2024b*). In *Bienertia sinuspersici*, recent work has identified molecular mechanisms central to $SCC_4$ photosynthesis, including the coexpression of genes associated with energy metabolism, cyclic electron flow, and $C_4$ transporters (*Han et al., 2023*). Additionally, differential antioxidant responses in peripheral and central chloroplasts suggest unique stress adaptation strategies of each chloroplast subtype (*Uzilday et al., 2023*). These findings highlight the agricultural potential

of SCC$_4$ mechanisms and establish a foundation for further exploring the genetic basis for SCC$_4$ systems.

Among SCC$_4$ species, *Bienertia sinuspersici* serves as an excellent model for investigating the C$_3$–SCC$_4$ transition due to the distinct developmental changes observed between its young and mature leaf stages. In the young, emergent leaves, chlorenchyma cells exhibit a typical C$_3$ photosynthetic phenotype, with chloroplasts dispersed throughout the cytoplasm and no intracellular compartmentalization or dimorphic chloroplasts. By contrast, in mature leaves, these cells undergo significant reorganization, developing into the specialized SCC$_4$ phenotype (*Offermann et al., 2015*). This includes the differentiation of dimorphic chloroplasts into central and peripheral types, the establishment of intracellular compartments, and the spatial localization of photosynthetic enzymes. Central chloroplasts, enriched with Rubisco and NAD-ME, focus on carbon fixation (analogous to bundle sheath cells), while peripheral chloroplasts, enriched with PEPC and CA, facilitate the initial steps of the C$_4$ pathway (analogous to mesophyll cells) (*Uzilday et al., 2023*). While significant progress has been made in characterizing the anatomy, protein distribution, and photosynthetic physiology of SCC$_4$ structural types in *Bienertia* (*Sharpe & Offermann, 2014*; *Offermann et al., 2015*; *Erlinghaeuser et al., 2016*; *Uzilday et al., 2023*), the molecular mechanisms driving these developmental transitions remain only partially understood. Recent transcriptomic studies, such as those conducted by *Han et al. (2023)*, have provided insights into the coexpression of SCC$_4$-related genes associated with energy metabolism, cyclic electron flow, and metabolite transport, highlighting their roles in coordinating photosynthesis across subcellular compartments. However, the precise molecular factors driving the establishment of dimorphic chloroplasts and the transition from C$_3$ to SCC$_4$ photosynthesis remain to be fully elucidated. To begin addressing this knowledge gap, we performed a comparative transcriptome analysis of emergent and mature leaves of *Bienertia sinuspersici*. By identifying key differentially expressed genes and enriched pathways, this study provides an initial framework for understanding the molecular basis of the C$_3$–SCC$_4$ transition, highlighting candidate genes that may play roles in dimorphic chloroplast development and SCC$_4$ functionality. While our findings capture snapshots of emergent and mature leaf states rather than the full transition, they serve as a foundation for future time-course studies tracking these molecular processes dynamically. Beyond advancing our understanding of *Bienertia*'s photosynthetic adaptation, such studies will have broader implications for improving crop productivity and resilience under environmental stress, such as heat and drought.

## MATERIALS & METHODS

### Plant material

*Bienertia sinuspersici* plants were maintained in 10-gallon (37.85-liter) citrus pots in growth chambers under a 14-hour light/10-hour dark cycle with a stepwise light regime increasing to 525 PPFM at full light and an 18 °C (dark) to 35 °C (light) temperature regime. Plants were watered once a week and were fertilized with Peters 20-21-5 in between waterings. Within two hours after light initiation, whole, fully expanded, mature leaves

and newly emerging young leaves (approximately 0.2 mm—File S1) were harvested from three 9-month-old vegetative stage plants, combined as a pooled sample, and immediately flash-frozen in liquid nitrogen. The timing of sample harvesting coincided with the fully photosynthetic stage of the leaf. Samples were pooled from three plants due to the limited tissue available for RNAseq. Flash-frozen leaf tissue was ground into a fine powder with a liquid nitrogen-cooled mortar and pestle. Approximately 100 mg of frozen powder was transferred to a liquid nitrogen-frozen 2 mL Eppendorf tube and stored at −80 °C until RNA was extracted. Portions of the text used in this section were previously published as part of a preprint (*Sharpe et al., 2023*).

## RNA extraction

Total RNA was extracted using an acid guanidinium thiocyanate phenol chloroform extraction method similar to that described previously (*Chomczynski & Sacchi, 1987*). Briefly, one mL of 0.8 M guanidinium thiocyanate, 0.4 M ammonium thiocyanate, 0.1 M sodium acetate pH 5.0, 5% w/v glycerol, and 38% v/v water saturated phenol were added to approximately 100 mg powdered tissue, shaken to evenly mix the sample, and incubated at room temperature for 5 min. 200 $\mu$L chloroform was added and shaken vigorously until the entire sample became uniformly cloudy before incubation at room temperature for 3 min. Samples were then centrifuged at 17,000× g at 4 °C for 15 min, and the aqueous phase was removed and transferred to a clean 1.5 mL Eppendorf tube. 600 $\mu$L 2-propanol was added, rocked 5 to 6 times, and incubated at room temperature for 10 min. Samples were centrifuged 17,000× g at 4 °C for 10 min, and the supernatant was discarded. one mL 75% DEPC-treated water mixed with ethanol was added, after which the samples were vortexed for 10 s and then centrifuged at 9,500× g at 4 °C for 5 min. The ethanol was removed by pipetting, and the pellets were allowed to dry completely. Pellets were suspended in RNase-free water and incubated at 37 °C with RNase-free DNase I (Thermo Scientific) for 30 min; the DNase I was inactivated at 65 °C for 10 min. 450 $\mu$L buffer RLC from the Qiagen (Valencia, CA, USA) RNeasy Plant Mini Kit was added to the digestion, processed in accordance with the manufacturer's recommendations, and eluted in 50 $\mu$L RNase free water. Extracted RNA was quality checked either with the Bio-Rad (Hercules, CA, USA) Experion system using the Experion RNA High Sens Analysis kit or the Agilent (Santa Clara, CA, USA) 2100 Bioanalyzer system using the RNA Nano Chip and Plant RNA Nano Assay Class.

## Illumina sequencing

cDNA and final sequencing library molecules were generated with Illumina's TruSeq RNA Sample Preparation v2 kit and instructions with minor modifications. Modifications to the published protocol include a decrease in the mRNA fragmentation incubation time from 8 min to 30 s to create the final library proper molecule size range. Additionally, A7Biosciences' (Woburn, MA, USA) DNA SizeSelector-I bead-based size selection system was utilized to target final library molecules for a mean size of 450 base pairs. All libraries were then quantified on a Life Technologies (Carlsbad, CA, USA) Qubit Fluorometer and qualified on an Agilent (Santa Clara, CA, USA) 2100 Bioanalyzer (Dr. Jeff Landgraf, personal

communication). The Illumina HiSeq 2000 sequencing platform was used to sequence 2x100 PE reads from the cDNA libraries generated from the above RNA extractions at Michigan State University's Research Technology Support Facility.

## 454 sequencing

cDNA libraries were constructed from the RNA extractions using the SMARTer™ PCR cDNA Synthesis Kit from ClonTech (Mountain View, CA, USA) according to the manufacturer's instructions. cDNA quality and size distribution were verified *via* 1% TAE gels and the Bio-Rad (Hercules, CA, USA) Experion system. cDNA libraries were processed to attach the Rapid Library Multiplex Identification (RL MID) Adapters according to the manufacturer's protocol. Libraries were then quality checked for size distribution with Agilent's (Santa Clara, CA, USA) 2100 Bioanalyzer, quantified *via* fluorometry, pooled, and then sequenced on Roche Applied Science's (Indianapolis, IN, USA) Genome Sequencer FLX System with GS FLX Titanium technology.

## Data availability

All sequencing data was submitted to the NCBI SRA and was assigned accession number PRJNA340188.

## Sequencing data QC and assembly

Sequence read information from Roche's GS FLX Standard Flowgram Format (sff) files included 70,867 reads for the mature dataset and 54,462 reads for the emergent dataset; read information from Illumina HiSeq 2000 2x100 PE fastq files included 178,716,218 reads for the mature dataset and 218,726,388 reads for the emergent dataset. All developmental read datasets were processed with the CLC Create Sequencing QC Report tool to assess read quality. The CLC Trim Sequence process was used to trim the 454 read datasets for a Phred value of 15; the Illumina reads were trimmed for a Phred score of 30, and the first twelve 5′bases were removed due to GC ratio variability. All read datasets were trimmed of ambiguous bases. Illumina reads were then processed through the CLC Merge Overlapping Pairs tool. In the absence of an available reference genome for *Bienertia sinuspersici*, we performed a *de novo* assembly using CLC Genomics Workbench v6 and according to methods described previously (*Hewitt, Ghogare & Dhingra, 2020*; *Sharpe et al., 2020*; *Hewitt et al., 2021*). Trimmed reads used for assembly were mapped back to the assembled contigs, mapped reads were used to update the contigs, and contigs with no mapped reads were ignored. Consensus contig sequences were filtered to include only those with a consensus length of >200 and a depth of coverage of >5. The resulting dataset was exported as a multi-fasta file. The individual mature and emergent leaf read datasets, from the original non-trimmed reads, were mapped back to the assembled contigs to generate individual developmental sample read counts for each contig; read counts were then normalized with the reads per kilobase per million reads (RPKM) method (*Mortazavi et al., 2008*).

## Functional annotation

The transcriptome fasta file produced from the assembly was imported into the OmicsBox Functional Annotation Module (Biobam Bioinformatics S.L., Valencia, Spain),

following which contig sequences were identified by a blastx alignment against the NCBI 'Viridiplantae' database, gene ontology (GO) mapping was performed, and annotations were assigned using the Blast2GO feature with default parameters (*Götz et al., 2008*).

### Differential expression analysis

Pairwise differential expression analyses were conducted for the emergent *versus* mature samples *via* the OmicsBox Transcriptomics Module using NOISeq-sim to compare the two developmental stages. NOISeq-sim infers significant differential expression without experimental replicates. For absent replicates, NOISeq-sim uses a multinomial distribution to model technical replicate read counts (*Tarazona et al., 2011*; *Tarazona et al., 2013*; *Tarazona et al., 2015*). Our research group has employed this method successfully in previous studies where it was not possible to obtain more than a single replicate per sample (*Hewitt et al., 2021*; *Hewitt et al., 2023*). *Bienertia sinuspersici* is difficult to sample in the wild, as it grows primarily in areas around the Persian Gulf. Since it is adapted to hot, dry, and high salt environments, it isn't easy to grow, even in climate-controlled chambers. Default parameters were used to simulate five replications with a set variability of 0.02 in each replication. Genes with a NOIseq probability greater than 0.9 and a |log2 fold change expression| value greater than 1.0 for at least one sample group were considered differentially expressed. A complete list of DE genes (along with the non-DE genes) and their corresponding functional annotations and expression values can be found in File S2.

### GO enrichment analysis

GO enrichment analysis using Fisher's exact test was conducted in the OmicsBox Functional Annotation Module to identify the cellular components, molecular functions, and biological processes that were enriched in each of the two developmental stages. Lists of the differentially expressed, functionally annotated genes were generated for the emergent and mature *Bienertia.* These lists served as the treatment datasets for enrichment analyses, and the master annotated transcriptome was used as the reference dataset. In addition to conducting a full GO enrichment analysis, a second analysis was run using the GO-Slim feature to reduce the number of GO terms present in the annotated reference transcriptome to include only general functions and processes displaying the greatest enrichment. For both analyses, an FDR-corrected *p*-value of 0.01 was used as the cutoff for determining statistical significance and to reduce the GO assignations to the most specific terms. Complete GO enrichment analysis and GO-slim analysis results can be found in File S3.

## RESULTS & DISCUSSION

### Gene expression and gene ontology enrichment analyses

*De novo* assembly yielded total of 116,257 contigs assembled from 141,504,502 trimmed reads with an N50 of 792 bases. Filtering for contigs with consensus length >200 and >5x coverage yielded 72,524 transcripts. Top blast hit descriptions were assigned to 26,007 contigs, and a subset of 24,603 contigs were fully annotated with corresponding gene ontologies and interpro IDs (File S2).

Expression analysis identified 3,364 DE genes, 1,761 in the emergent tissue and 1,603 genes in the mature tissue with high probability of differential expression. Of these,

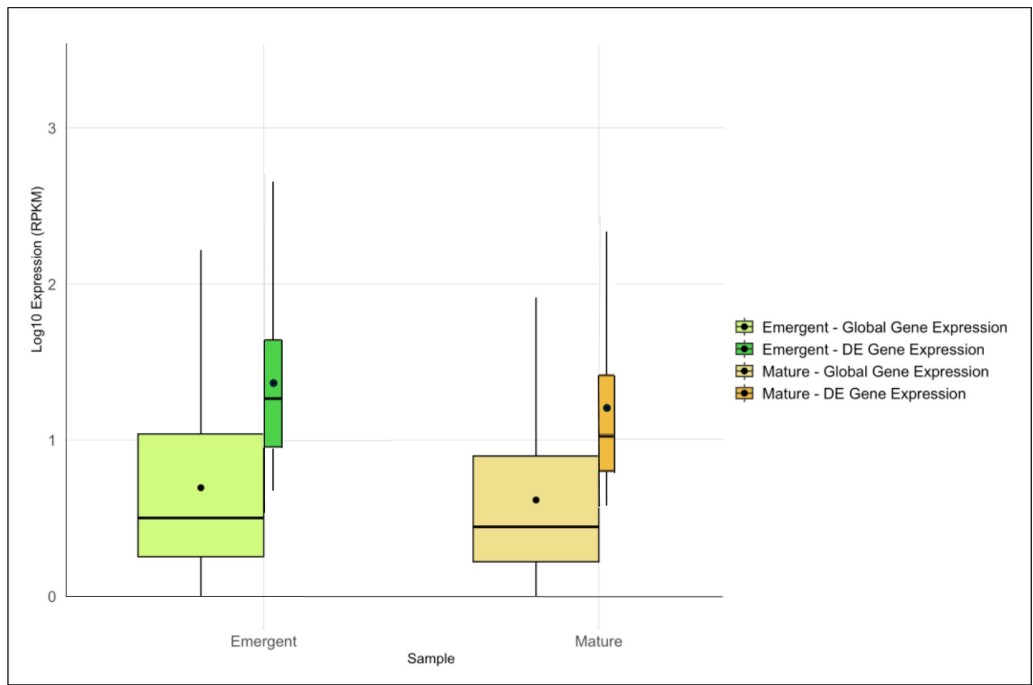

**Figure 1** **Overview of global and differentially expressed (DE) gene expression profiles in emergent *versus* mature *Bienertia sinuspersici* leaves.** The box plots compare the log10-transformed RPKM values for each sample, and each stage is represented by two adjacent box plots representing global gene expression and DE gene expression. The widths of the box plots are proportional to the number of genes. The median and mean expression levels are indicated by central lines and dots, respectively.

1,062 and 661 DE genes were matched with functional annotations in the emergent and mature tissues, respectively (File S2). Emergent leaves exhibited slightly higher median and mean global and DE gene expression levels than mature leaves, as reflected in the log10-transformed RPKM values (Fig. 1). Additionally, a broader range of expression values was observed in the emergent leaves compared to the mature leaves, suggesting more diverse transcriptional activity in the former. This observation may reflect a broader range of cellular activities required for early leaf development and cellular differentiation.

The assignment of GO terms to annotated datasets enabled the identification and classification of biological processes, molecular functions, and cellular components that were overrepresented in each of the treatment datasets *versus* the entire transcriptome. A total of 110 GOs associated with biological processes (bpGOs), 43 GOs associated with molecular functions (mfGOs), and 21 GOs associated with cellular components (ccGOs) were uniquely overrepresented in the emergent tissues; 48 bpGOs, 23 mfGOs, and six ccGOs were uniquely overrepresented in the mature tissues; and 27 bpGOs, 11 mfGOs, and 10 ccGOs were overrepresented in both emergent and mature tissues (File S3). To further simplify the dataset and identify the more inclusive ontologies representative of each developmental stage, we used the OmicsBox GO-slim tool to create a reduced list of enriched ontologies. The simplified ontology list was comprised of 10 bpGOs, four mfGOs, and four ccGOs uniquely enriched in the emergent tissue; two bpGOs, two mfGO, and

one ccGO uniquely enriched in the mature tissue; and 20 bpGOs, seven mfGOs, and eight ccGOs that were enriched in both emergent and mature tissues (File S3).

GO-slim results provided a broad overview of enriched functions, guiding the analysis by highlighting key pathways. More specific GO terms (from the comprehensive enrichment analysis) corresponding to the GO-slims offer detailed insight into the enriched processes at each developmental stage (Table 1). In emergent tissues, unique enrichment was observed in terms related to cell cycle, growth, and differentiation, including specific GOs such as 'cyclin-dependent protein serine/threonine kinase activity', 'regulation of G2/M transition of mitotic cell cycle', 'regulation of cell division', 'regulation of cell growth/size', 'asymmetric cell division', and 'regulation of cell population proliferation'. Terms associated with cytoskeleton and cytoskeletal motor activities were also uniquely enriched in the emergent stage, with specific GOs including 'cortical microtubule', 'microtubule-based movement', 'spindle assembly', and 'phragmoplast'. These results highlight the stage-specific importance of processes essential for cell division, structural establishment, and spatial organization, which are necessary for preparing emergent tissues for later specialization.

In contrast, mature tissues exhibited unique enrichment of GOs associated with photosynthetic processes, including specific terms such as 'chloroplast thylakoid membrane', 'chlorophyll binding', 'photosystem II, light harvesting', and 'photosynthetic electron transport chain'. Terms related to homeostasis and oxygen-binding processes, such as 'oxygen binding' and 'peroxidase activity', were also uniquely enriched, reflecting the tissue's transition to specialized photosynthetic functions and adaptations for maintaining internal stability. Additionally, carbohydrate-related terms, such as 'carbohydrate binding' and 'sucrose-phosphate synthase activity', highlight the importance of resource management and structural integrity in photosynthetically active tissue.

GO terms that were highly enriched at both developmental stages included terms associated with transcription-mediated response to stimuli—*e.g.*, 'DNA binding', 'DNA-binding transcription factor activity', and 'regulation of DNA-templated transcription'—and cellular component organization—*e.g.*, 'system development', 'cytoplasmic vesicle', 'vacuole', 'mitochondria', and 'chloroplast stroma'. The shared enrichment of these GOs highlights core processes critical for transcriptional regulation and organelle organization at both stages of development.

Analysis of key DE genes corresponding to enriched pathways provides further insights into the developmental processes driving SCC$_4$ functionality in *Bienertia*. The following sections examine key genes associated with the enriched GO terms, highlighting the roles these genes play as tissues transition from growth and differentiation in the emergent stage to functional specialization in the mature stage.

## Genes associated with the cell cycle and growth

In early leaf development, meristematic cells actively divide; however, as leaves mature, cell proliferation becomes confined to the leaf base while cells elongate and expand (*Gonzalez, Vanhaeren & Inzé, 2012*; *Kalve, De Vos & Beemster, 2014*). Underpinning these developmental processes, the cell cycle is regulated by cyclins (CYCs), cyclin-dependent

**Table 1** Enriched GO-slim ontologies and the corresponding specific ontologies from the comprehensive GO analysis (FDR corrected *p*-value < 0.01).

| GO-slim terms | Specific GO terms |
|---|---|
| **EMERGENT YOUNG LEAF TISSUE** | |
| **Cytoskeletal motor activity, Cytoskeleton** | Condensin complex, Cortical microtubule, Condensin complex, Cortical microtubule, Cyclin-dependent protein serine/threonine kinase activity, Cyclin-dependent protein serine/threonine kinase regulator activity, Kinesin complex, Microtubule binding, Microtubule-based movement, Minus-end-directed microtubule motor activity, Phragmoplast, Preprophase band, Spindle |
| **Cell cycle** **Cell differentiation** **Cell growth** | Asymmetric cell division, Cell plate formation during cytokinesis, Cell wall modification, Chromatin binding, Cyclin-dependent protein kinase holoenzyme complex, Cyclin-dependent protein serine/threonine kinase activity, DNA helicase activity, DNA methylation-dependent heterochromatin formation, DNA replication initiation, Endoplasmic reticulum organization, Histone H3S10 kinase activity, Lateral root formation, Leaf development, Microtubule polymerization, Multidimensional cell growth, Nucleosome assembly, PCNA complex, Plant-type cell wall biogenesis, Plant-type cell wall organization and biogenesis, Regionalization, Regulation of anatomical structure morphogenesis, Regulation of cell division, Regulation of cell growth, Regulation of cell population proliferation, Regulation of cell size, Regulation of DNA replication, Regulation of flower development, Regulation of G2/M transition of the mitotic cell cycle, Regulation of meristem growth, Spindle assembly, Stomatal complex morphogenesis |
| **MATURE LEAF TISSUE** | |
| **Photosynthesis** **Thylakoid** | Chlorophyll binding, Chloroplast membrane, Chloroplast organization, Chloroplast thylakoid membrane, Light-harvesting complex, MAPK cascade, Nonphotochemical quenching, Pentose-phosphate shunt, Photosynthesis, light harvesting, Photosynthetic electron transport chain, Photosystem I reaction center, Photosystem II, Photosystem II oxygen-evolving complex assembly, Pigment biosynthetic process, Response to blue light, red light, far-red light, Thylakoid membrane organization |
| **Oxygen binding** | 4 iron, 4 sulfur cluster binding, Oxidoreductase activity, acting on paired donors, Oxygen binding, Peroxidase activity |
| **Carbohydrate binding** | Beta-amylase activity, Monosaccharide binding, Polysaccharide binding, Response to sucrose, Sucrose-phosphate synthase activity |
| **Cellular homeostasis** | Cell redox homeostasis, Hydrogen peroxide metabolic process, Intracellular monoatomic cation homeostasis, Response to gibberellin, Response to iron ion starvation, Response to nitrate, Response to oxidative stress, Systemic acquired resistance, salicylic acid-mediated signaling pathway |
| **SHARED - EMERGENT YOUNG & MATURE LEAF TISSUE** | |
| **Cellular component organization** | System development, cytoplasmic vesicle, vacuole, mitochondrion, nucleus, chloroplast stroma, cell wall, cytosol, organelle envelope |
| **Response to stimulus** | DNA binding, DNA-binding transcription factor activity, regulation of DNA-templated transcription |

kinases (CDKs), aurora kinases, condensin complexes, and mitotic spindle checkpoints, which collectively govern the transitions between cell cycle phases (*Collette et al., 2011*; *Willems et al., 2018*; *Shimotohno et al., 2021*; *Li et al., 2024a*; *Li et al., 2024b*).

Aligning with this established information, in *Bienertia*, the largest group of enriched GO terms identified in the emergent tissue were associated with cell cycle regulation, cell growth, and cellular differentiation. Analysis of the DE genes corresponding to these enriched GOs revealed differential expression of genes corresponding to various components of the cell cycle, particularly genes associated with cellular duplication. Transcripts corresponding to *CDKB1-1*, along with *CYCA1*, *CYCA2*, *CYCA3-1*, *CYCD2*, *CYCD3*, *CYCA1-like*, and *CYCS13-6* were among those that were significantly upregulated in the emergent tissue (Table 2); all of these are known to accumulate during the mitotic

**Table 2 Differentially expressed genes associated with cell cycle, growth, and differentiation; cytoskeleton and cytoskeletal motor activity; and photosynthetic and carbon fixation-related processes in emergent (E) and mature (M)** *Bienertia sinuspersici* **leaf tissue.** The log2FC (M/E) expression values corresponding to genes that were significantly upregulated in the young tissues (value < −1) are italicized, and those corresponding to genes that were significantly upregulated in the mature tissues (value < 1) are in bold.

| Contig # | Gene | Associated general processes (GO-Slim) | Emergent RPKM | Mature RPKM | Log2FC (M/E) |
|---|---|---|---|---|---|
| 26433 | AUR1 | Cell cycle/cell growth/differentiation | 28.76 | 4.01 | *−2.84* |
| 26432 | AUR1 | Cell cycle/cell growth/differentiation | 27.28 | 3.91 | *−2.80* |
| 41110 | AUR2 | Cell cycle/cell growth/differentiation | 47.22 | 7.13 | *−2.73* |
| 24169 | AUR3 | Cell cycle/cell growth/differentiation | 11.01 | 1.67 | *−2.72* |
| 14854 | CDKB1-1 | Cell cycle/cell growth/differentiation | 47.13 | 5.78 | *−3.03* |
| 15194 | CDKF4 | Cell cycle/cell growth/differentiation | 0.08 | 9.43 | **6.88** |
| 19543 | CLV1 | Cell cycle/cell growth/differentiation | 2.39 | 8 | **1.74** |
| 8415 | CYC1-like | Cell cycle/cell growth/differentiation | 30.89 | 4.05 | *−2.93* |
| 35088 | CYCA1 | Cell cycle/cell growth/differentiation | 26.2 | 3.67 | *−2.84* |
| 39391 | CYCA2 | Cell cycle/cell growth/differentiation | 25.48 | 3.84 | *−2.73* |
| 14582 | CYCA3-1 | Cell cycle/cell growth/differentiation | 10 | 1.82 | *−2.46* |
| 41850 | CYCB | Cell cycle/cell growth/differentiation | 47.18 | 7.57 | *−2.64* |
| 49181 | CYCB1 | Cell cycle/cell growth/differentiation | 23.15 | 3.93 | *−2.56* |
| 37875 | CYCD2 | Cell cycle/cell growth/differentiation | 18.38 | 2.92 | *−2.65* |
| 10418 | CYCD3 | Cell cycle/cell growth/differentiation | 141.66 | 19.25 | *−2.88* |
| 41309 | CYCS13-6 | Cell cycle/cell growth/differentiation | 46.52 | 7.58 | *−2.62* |
| 60415 | CYCS13-6 | Cell cycle/cell growth/differentiation | 23.01 | 3.82 | *−2.59* |
| 25901 | ERECTA | Cell cycle/cell growth/differentiation | 87.81 | 11.52 | *−2.93* |
| 3912 | ERL | Cell cycle/cell growth/differentiation | 67.72 | 8.02 | *−3.08* |
| 19390 | MAD2 | Cell cycle/cell growth/differentiation | 25.11 | 2.99 | *−3.07* |
| 34358 | MAD3 | Cell cycle/cell growth/differentiation | 8.36 | 1.36 | *−2.62* |
| 3636 | MAPKKK 1-like | Cell cycle/cell growth/differentiation | 1.12 | 4.1 | **1.87** |
| 24473 | PXL2-like | Cell cycle/cell growth/differentiation | 3.46 | 13.97 | **2.01** |
| 13131 | SMC1 | Cell cycle/cell growth/differentiation | 26.45 | 5.32 | *−2.31* |
| 24311 | SMC2 | Cell cycle/cell growth/differentiation | 16.77 | 3.22 | *−2.38* |
| 16156 | SMC3 | Cell cycle/cell growth/differentiation | 18.26 | 2.87 | *−2.67* |
| 16157 | SMC3 | Cell cycle/cell growth/differentiation | 10.73 | 2.04 | *−2.40* |
| 30868 | WAKL2-like | Cell cycle/cell growth/differentiation | 2.51 | 11.46 | **2.19** |
| 53995 | WAKL20 | Cell cycle/cell growth/differentiation | 7.19 | 1.29 | *−2.48* |
| 11102 | WAKL4 | Cell cycle/cell growth/differentiation | 2.85 | 11.77 | **2.05** |
| 28646 | WAKL8-like | Cell cycle/cell growth/differentiation | 1.25 | 4.75 | **1.93** |
| 28645 | WAKL9-like | Cell cycle/cell growth/differentiation | 1.2 | 5.36 | **2.16** |
| 167 | ACT | Cytoskeleton | 570.49 | 205.31 | *−1.47* |
| 6918 | KIF-like | Cytoskeleton | 27.37 | 4.8 | *−2.51* |
| 20555 | KIF-like | Cytoskeleton | 11.85 | 2.27 | *−2.38* |
| 37370 | KIF-like | Cytoskeleton | 10.94 | 0.37 | *−4.89* |
| 53507 | KIF | Cytoskeleton | 10.74 | 1.92 | *−2.48* |

*(continued on next page)*

**Table 2** (*continued*)

| Contig # | Gene | Associated general processes (GO-Slim) | Emergent RPKM | Mature RPKM | Log2FC (M/E) |
|---|---|---|---|---|---|
| 25323 | KIF | Cytoskeleton | 10.31 | 1.46 | −2.82 |
| 1483 | KIF | Cytoskeleton | 19.24 | 2.88 | −2.74 |
| 15415 | KIF | Cytoskeleton | 32.93 | 4.63 | −2.83 |
| 19882 | KIF | Cytoskeleton | 18.54 | 2.59 | −2.84 |
| 21707 | KIF | Cytoskeleton | 39.73 | 5.55 | −2.84 |
| 22027 | KIF | Cytoskeleton | 7.59 | 1.26 | −2.59 |
| 46667 | KIF | Cytoskeleton | 5.18 | 0.61 | −3.09 |
| 20861 | KIF-like NACK1 | Cytoskeleton | 19.89 | 2.7 | −2.88 |
| 29212 | KIF18B | Cytoskeleton | 9.43 | 1.63 | −2.53 |
| 66942 | KIF18B | Cytoskeleton | 8.61 | 0.35 | −4.62 |
| 34672 | KIF18b-like | Cytoskeleton | 13.62 | 1.48 | −3.20 |
| 3559 | KIF2 | Cytoskeleton | 40.8 | 6.89 | −2.57 |
| 2774 | KIF2 | Cytoskeleton | 40.16 | 6.3 | −2.67 |
| 16133 | KIF22-like | Cytoskeleton | 23.85 | 3.95 | −2.59 |
| 19896 | KIF4A | Cytoskeleton | 7.44 | 1.15 | −2.69 |
| 32317 | MAP | Cytoskeleton | 9.64 | 1.93 | −2.32 |
| 8446 | MAP65-3 | Cytoskeleton | 22.82 | 4.12 | −2.47 |
| 20026 | MAP65-3 | Cytoskeleton | 16.23 | 2.92 | −2.47 |
| 73731 | MAP65-3 | Cytoskeleton | 6.98 | 0.89 | −2.97 |
| 23646 | MAP65-5 | Cytoskeleton | 17.54 | 3.03 | −2.53 |
| 17227 | MAP RE1-2 | Cytoskeleton | 19.32 | 2.46 | −2.97 |
| 22548 | MAP RE1-2 | Cytoskeleton | 12.76 | 2.58 | −2.31 |
| 17934 | TUBA | Cytoskeleton | 186.36 | 31.89 | −2.55 |
| 2771 | TUBA | Cytoskeleton | 177.04 | 28.59 | −2.63 |
| 9626 | TUBB3 | Cytoskeleton | 247.53 | 52.97 | −2.22 |
| 3756 | CA | Photosynthesis/Carbon fixation | 27.68 | 146.62 | **2.41** |
| 3756 | CA | Photosynthesis/Carbon fixation | 27.68 | 146.62 | **2.41** |
| 10353 | γ-CA | Photosynthesis/Carbon fixation | 124.02 | 27.57 | −2.17 |
| 6660 | FeSOD | Photosynthesis/Carbon fixation | 69.95 | 237.83 | **1.77** |
| 169 | LCHB | Photosynthesis/Carbon fixation | 2975.1 | 9136.22 | **1.62** |
| 34 | LHCB-like | Photosynthesis/Carbon fixation | 1505.8 | 5130.86 | **1.77** |
| 3123 | LHCB6 | Photosynthesis/Carbon fixation | 180.73 | 474.78 | **1.39** |
| 5116 | LIL3 | Photosynthesis/Carbon fixation | 206.24 | 43.94 | −2.23 |
| 1005 | MgPME cyclase | Photosynthesis/Carbon fixation | 198.14 | 647.47 | **1.71** |
| 39291 | OEP80 | Photosynthesis/Carbon fixation | 29.41 | 6.19 | −2.25 |
| 3006 | PEPC | Photosynthesis/Carbon fixation | 43.69 | 125.88 | **1.53** |
| 22450 | PEPC | Photosynthesis/Carbon fixation | 26.78 | 1.44 | −4.22 |
| 41291 | PP2C | Photosynthesis/Carbon fixation | 10.35 | 1.97 | −2.39 |
| 11856 | PsaC | Photosynthesis/Carbon fixation | 1.52 | 5.28 | **1.80** |
| 198 | PsaE | Photosynthesis/Carbon fixation | 311.53 | 1295.95 | **2.06** |
| 266 | PsaH | Photosynthesis/Carbon fixation | 1129.81 | 3158.12 | **1.48** |
| 721 | PsaI | Photosynthesis/Carbon fixation | 753.25 | 2097.68 | **1.48** |

**Table 2** (*continued*)

| Contig # | Gene | Associated general processes (GO-Slim) | Emergent RPKM | Mature RPKM | Log2FC (M/E) |
|---|---|---|---|---|---|
| 599 | PsaK | Photosynthesis/Carbon fixation | 674.62 | 2059.64 | **1.61** |
| 771 | PsbY | Photosynthesis/Carbon fixation | 222.42 | 578.99 | **1.38** |
| 18279 | PSII D1 | Photosynthesis/Carbon fixation | 3.17 | 14.73 | **2.22** |
| 819 | RIP1 precursor | Photosynthesis/Carbon fixation | 26.32 | 430.38 | **4.03** |
| 11911 | RPO-D | Photosynthesis/Carbon fixation | 13 | 39.26 | **1.59** |
| 102 | RUBISCO small subunit | Photosynthesis/Carbon fixation | 3445.83 | 11105.92 | **1.69** |
| 29496 | RUBISCO small subunit | Photosynthesis/Carbon fixation | 73.39 | 222.16 | **1.60** |
| 21034 | SIG | Photosynthesis/Carbon fixation | 4.22 | 14.71 | **1.80** |
| 9253 | TOC75 | Photosynthesis/Carbon fixation | 154.5 | 11.4 | −3.76 |

synthesis phase (S), the second gap (G2) phase, and the mitosis (M) phase (*Boudolf et al., 2004*; *Boudolf et al., 2009*; *Inzé & De Veylder, 2006*). Only one cyclin-associated transcript, *CDKF4*, was differentially expressed in mature tissue. Along with *CYC/CDK*, the emergent tissue displayed significantly increased expression of transcripts corresponding to serine-threonine protein kinases *AURORA1*, *AURORA2*, *AURORA3* (*AUR1*, *AUR2*, *AUR3*); *CONDENSIN COMPLEX SUBUNIT 1*, *2*, and *3* (*SMC1*, *SMC2*, and *SMC3*); and mitotic spindle checkpoint-associated *MITOTIC ARREST DEFICIENCY 2* and *3* (*MAD2* and *MAD3*). *AUR* and *SMC* are important mediators of cellular mitosis, with crucial roles in G2/M transition, chromosome binding, and kinetochore separation (*Collette et al., 2011*; *Willems et al., 2018*).

In addition to core cell cycle regulators, the developmental patterning of leaf tissue can be monitored *via* the expression of leucine-rich repeat receptor-like protein kinases (LRR-RLK) family members. These proteins detect and transduce signals to initiate responses critical for the development of shoot organs. One clade of the LRR-RLK family, including ERECTA and ERECTA-like (ERL), localizes and functions in the shoot apical meristem (SAM) and organ primordia. Previous work has demonstrated that *ERECTA* mRNA is expressed at low levels in the SAM, with expression increasing in developing vegetative organs and decreasing in mature organs (*Yokoyama et al., 1998*). Consistent with this, *Bienertia* emergent leaf tissue displayed significantly higher expression levels of transcripts corresponding to *ERECTA* and *ERL* compared to mature tissues (Table 2). Not all LRR-RLK genes are developmentally expressed to promote actively dividing cells in emergent tissues; some, such as *PHLOEM INTERCALATED WITH XYLEM 2-like* (*PXL2-like*) and wall-associated receptor kinase *CLAVATA1* (*CLV1*), act in maturing tissues to suppress cell division or generate function-specific cell types in expanding tissues. *PXL2-like* has been implicated in the development of phloem and xylem (*Etchells et al., 2013*), while *CLV1* facilitates phyllotaxis formation in leaf primordia and suppresses undifferentiated cells at the shoot meristem, committing them to organ development (*Clark, Running & Meyerowitz, 1993*). These genes, significantly upregulated in mature tissues, suggest an active role in vascular differentiation and morphological organization during the later stages of leaf development. Beyond the LRR-RLK family, wall-associated kinases (WAKLs)

contribute to the development of leaf structure and morphology during development. Several *WAKL* genes (*WAKL2*, *WAKL4*, *WAKL8*, and *WAKL9*), which are known to mediate cellular signaling and wall dynamics, were highly expressed in mature tissues (*Verica et al., 2003*; *Sharma et al., 2020*). Together, these findings support the structural and physiological adaptations necessary for functional specialization in mature leaves.

## Genes involved in cytoskeletal and structural organization

Cytoskeletal dynamics are central to the coordination of cell division and structural development, providing stability and facilitating key processes such as mitotic spindle formation, organelle positioning, and cytoplasmic streaming (*Smertenko et al., 2018*). The cytoskeleton consists of actin filaments and microtubules, each serving distinct functions: actin primarily drives cytoplasmic streaming, while microtubules support cell wall assembly and intracellular transport. Microtubule-associated proteins (MAPs), including motor MAPs like kinesins and non-motor MAPs that stabilize and regulate microtubule dynamics, are crucial for these functions (*Parrotta, Cresti & Cai, 2014*; *Lee, Qiu & Liu, 2015*). In *Bienertia*, cytoskeletal components are particularly relevant to the development of dimorphic chloroplasts, as the chloroplasts of *Bienertia* species have been shown to interact with microtubules and actin as the chlorenchyma cell develops (*Voznesenskaya et al., 2005*; *Chuong, Franceschi & Edwards, 2006*) (Table 1).

Nearly all differentially expressed cytoskeleton-associated genes were more highly expressed in emergent leaf tissues, highlighting the critical role of cytoskeletal components in cell division, structural establishment, and intracellular organization during early leaf development. *β-ACTIN* (*ACTB*), a key component of actin filaments, was highly expressed at both developmental stages with significant differential expression in emergent tissues. The comparatively high expression of *ACTB* in early development may reflect its importance for organelle positioning and the establishment of foundations for further compartmentalization. Notable among the motor MAPs expressed in emergent tissues are kinesin-like proteins (*KIFs*), such as *KIF18B*, *KIF4A*, and *KIF22*, which are involved in spindle dynamics, chromosome alignment, and intracellular transport, processes integral to cell division and structural organization (*Lee, Qiu & Liu, 2015*). Additionally, non-motor MAP genes such as *MAP65-3* and *MAP65-5* contribute to microtubule bundling and stabilization during mitosis, facilitating cytokinesis and cell wall assembly (*Parrotta, Cresti & Cai, 2014*). *α-TUBULIN* (*TUBA*) and *β-TUBULIN3* (*TUBB3*) were similarly expressed, highlighting their involvement in spindle formation and intracellular movement during cell proliferation. Together, these findings highlight the important role of microtubule dynamics, not only in establishing cellular architecture but also in enabling the structural and functional transitions unique *to Bienertia*.

## Genes associated with photosynthetic development

Photosynthetic development in *Bienertia* is governed by transcriptional programs that drive chloroplast biogenesis, maturation, and the establishment of dimorphic chloroplast functionality required for $SCC_4$ carbon fixation. Most photosynthesis and carbon fixation-related genes identified in this study showed differential expression in mature tissues

(Table 1), consistent with the enriched GO category for photosynthesis, while genes in emergent tissues appeared to establish foundational processes.

In mature tissues, genes encoding light-harvesting complex proteins, such as *LIGHT HARVESTING CHLOROPHYLL-BINDING* (*LCHB*), *LCHB-like*, and *LCHB6*, facilitate energy capture and transfer to the photosystems, critical for sustaining photosynthesis. Genes encoding subunits of PHOTOSYSTEM I (PSI)—including *PSA-C*, *PSA-E*, *PSA-H*, *PSA-I*, and *PSA-K*—were more highly expressed in mature tissues, highlighting their roles in electron transfer and energy conversion at this stage (*Jensen et al., 2000*; *Jensen et al., 2007*). Regarding PHOTOSYSTEM II (PSII), the D1 PROTEIN encoding gene (*PSII D1*), a core component involved in water splitting and oxygen evolution, and *PSB-Y*, a stabilizer of the PSII complex, were also strongly expressed. The mature tissue-specific expression of *IRON SUPEROXIDE DISMUTASE* (*FeSOD*) likely contributes to redox balance within chloroplasts, mitigating oxidative stress generated during high rates of photosynthesis. Regulatory and biosynthetic processes supporting chloroplast functionality were evident in mature tissues. *RNA POLYMERASE SIGMA FACTOR D* (*RPO-D*) and *SIGMA FACTOR* (*SIG*) were expressed, potentially regulating chloroplast gene expression. *MAGNESIUM-PROTOPORPHYRIN IX MONOMETHYL ESTER CYCLOHYDROLASE* (*MgPME cyclase*), which supports chlorophyll biosynthesis, was also highly expressed in mature tissues, potentially reflecting an increased demand for chlorophyll to sustain higher rates of photosynthesis in mature leaves (*Takahashi et al., 2014*; *Kong et al., 2016*).

While photosynthetic gene expression in emergent tissues was generally lower than in mature tissues, several genes essential for early chloroplast biogenesis and organization that were expressed at the emergent stage included *LIGHT-INDUCED PROTEIN 3* (*LIL3*), which supports chlorophyll production by stabilizing biosynthetic intermediates (*Takahashi et al., 2014*). Structural assembly and functional development of emergent chloroplasts appears to be further supported by *OUTER ENVELOPE PROTEIN 80* (*OEP80*) and *TRANSLOCON OF THE OUTER CHLOROPLAST MEMBRANE 75* (*TOC75*), which mediate the import of nuclear-encoded proteins into chloroplasts (*Yoshimura et al., 2024*). These genes may play critical roles in establishing the distinct structural and functional identities of central and peripheral chloroplasts in *Bienertia*.

With regards to genes associated with carbon fixation, high expression of *CA* was observed in mature tissues, reflecting its role in supporting efficient carbon fixation within peripheral chloroplasts as $SCC_4$ photosynthesis becomes established. Additionally, *PEPC* was expressed at both stages, with one transcript more highly expressed in emergent tissues and another in mature tissues. This suggests that the differential expression of *PEPC* transcripts may play a role in supporting the establishment and specialization of distinct chloroplast functions during the transition from $C_3$ to $SCC_4$ photosynthesis. Specifically, these expression patterns could reflect the progressive development of the spatially separated roles of peripheral and central chloroplasts in the $SCC_4$ pathway. Finally, the elevated expression of two transcripts corresponding to the *RUBISCO* small subunit in mature tissues reflects its increased role in central chloroplasts during the mature stage, when carbon fixation becomes fully operational. This pattern suggests a

developmental shift in chloroplast function to support the heightened demands of $SCC_4$ photosynthesis.

## Differentially expressed transcription factors

Among the GOs enriched at both emergent and mature stages were several associated with transcriptional regulation. Given the central role of transcription factors (TFs) in orchestrating developmental processes, stage-specific expression of TFs in *Bienertia* was explored (Table 3).

In emergent tissues, a higher diversity and abundance of transcription factor-associated genes were observed, including genes encoding auxin response factor (ARF), basic helix-loop-helix (bHLH), homeodomain (HD), MYB, squamosa promoter-binding protein-like (SPL), and zinc finger (ZF) family proteins. These TFs likely regulate gene expression programs critical for cell proliferation, differentiation, and early leaf anatomical development. For example, *ARF* and *MYB* transcription factor genes, which are implicated in hormonal signaling pathways such as auxin and gibberellin signaling (*Shin et al., 2007*; *Zhao et al., 2021*), were highly expressed in emergent tissues, suggesting a role in the coordination of hormone responses necessary for growth and the establishment of leaf structures. Similarly, *NAC* and *SPL* transcription factor genes, known to mediate stress responses and metabolic regulation (*Filichkin et al., 2018*; *Hernandez, Goswami & Sanan-Mishra, 2020*; *Min et al., 2022*), likely contribute to early adaptation mechanisms ensuring survival and proper development during this critical stage, while *BEL1-like*, a homeodomain-containing TF, functions in the shoot apex to regulate meristem identity and facilitate the correct development of shoot architecture (*Kumar et al., 2007*).

The heightened expression of *bHLH* transcription factors, such as *TRANSCRIPTION FACTOR SPEECHLESS-LIKE*, at the emergent stage, aligns with GO terms related to cell cycle regulation and cytoskeletal dynamics, pointing to their involvement in stomatal developmental progression and structural organization (*Chen, Wu & Hou, 2020*). Additionally, *GATA* TF genes, with known roles in regulating chloroplast biogenesis (*Chiang et al., 2012*; *Hudson et al., 2013*), may lay the groundwork during the emergent stage for the transition to photosynthetic functionality.

In mature tissues, fewer TF families were differentially expressed, but a distinct enrichment of highly expressed ethylene-responsive transcription factor genes (*ERFs*) was observed. These *ERFs*, including *ERF1A*, *ERF2*, and *DEHYDRATION-RESPONSIVE ELEMENT BINDING PROTEIN*, are commonly associated with stress responses, senescence, and hormone-mediated cross-talk (*Nakano et al., 2006*). The prominence of *ERF* genes suggests their regulatory roles in maintaining cellular homeostasis and responding to environmental stimuli during the mature photosynthetic stage. Additionally, the mature stage showed expression of *CONSTANS* (*CO*) and *CONSTANS-like* (*COL*), zinc finger TFs linked to photoperiodic regulation (*Romero et al., 2024*), which aligns with the enriched GO terms for photosynthesis and light response.

## Framework for $SCC_4$ and chloroplast differentiation

The findings of our transcriptomic analysis provide a framework for understanding how stage-specific gene expression contributes to the structural and functional transitions

**Table 3  Differentially expressed, transcription factor-encoding genes in emergent (E) and mature (M) *Bienertia sinuspersici* leaf tissue.** The log2FC (M/E) expression values corresponding to genes that were significantly upregulated in the young tissues (value < −1) are italicized, and those corresponding to genes that were significantly upregulated in the mature tissues (value < 1) are in bold.

| Contig # | Gene name | TF class | Emergent RPKM | Mature RPKM | Log2FC (M/E) |
|---|---|---|---|---|---|
| 87 | Auxin response factor 3-like | ARF | 22.68 | 3.29 | *−2.79* |
| 16663 | Auxin response factor 4 | ARF | 30.09 | 5.02 | *−2.58* |
| 16662 | Auxin response factor 4 | ARF | 2.81 | 15.17 | **2.43** |
| 33726 | Auxin response factor 4-like | ARF | 16.12 | 2.75 | *−2.55* |
| 37973 | bHLH domain-containing protein | bHLH | 10.96 | 2.16 | *−2.34* |
| 3634 | Transcription factor bHLH 145-like | bHLH | 43.85 | 6.04 | *−2.86* |
| 31753 | Transcription factor bHLH66 | bHLH | 6.2 | 1.17 | *−2.41* |
| 47610 | Transcription factor bHLH93 | bHLH | 22.78 | 4.51 | *−2.34* |
| 13021 | Transcription factor bHLH96-like | bHLH | 51.6 | 11.51 | *−2.16* |
| 21892 | Transcription factor ORG2-like | bHLH | 9.4 | 1.61 | *−2.55* |
| 27143 | Transcription factor SPEECHLESS-like | bHLH | 13.74 | 1.04 | *−3.72* |
| 13125 | Transcription factor UNE10-like | bHLH | 12.75 | 37.22 | **1.55** |
| 10322 | Dehydration-responsive element binding protein | ERF | 22.94 | 96.89 | **2.08** |
| 52853 | Ethylene responsive transcription factor ERF098-like | ERF | 0.28 | 4.12 | **3.88** |
| 26140 | Ethylene responsive transcription factor ERF1A | ERF | 2.63 | 10.13 | **1.95** |
| 28362 | Ethylene responsive transcription factor ERF1b-like | ERF | 1.18 | 6.07 | **2.36** |
| 46784 | Ethylene responsive transcription factor ERF2 | ERF | 2.22 | 7.92 | **1.83** |
| 83348 | Ethylene-responsive transcription factor | ERF | 12.51 | 1.62 | *−2.95* |
| 72889 | Ethylene-responsive transcription factor ERF017-like | ERF | 25.84 | 0.71 | *−5.19* |
| 57854 | GATA transcription factor | GATA | 12.49 | 2.16 | *−2.53* |
| 54258 | GATA transcription factor 18-like | GATA | 4.76 | 0.81 | *−2.55* |
| 31670 | BEL1-like transcription factor | HD | 27.64 | 4.55 | *−2.60* |
| 21136 | Homeobox-leucine zipper protein ATHB-12-like | HD | 4.73 | 15.42 | **1.70** |
| 49851 | Homeobox-leucine zipper protein ATHB-40-like | HD | 15.76 | 1 | *−3.98* |
| 25288 | Homeobox-leucine zipper protein ATHB-52 | HD | 0.06 | 3.75 | **5.97** |
| 25421 | Homeobox-leucine zipper protein ATHB-6-like | HD | 9.3 | 1.74 | *−2.42* |
| 22718 | Homeobox-leucine zipper protein HDG2-like | HD | 51.85 | 9.66 | *−2.42* |
| 20552 | Homeobox-leucine zipper protein ROC3-like | HD | 18.95 | 3.1 | *−2.61* |
| 27094 | Wuschel-related homeobox 1-like | HD | 4.84 | 0.74 | *−2.71* |
| 57231 | MYB transcription factor | MYB | 5.78 | 1.02 | *−2.50* |
| 32314 | MYB transcription factor r2r3 | MYB | 6.44 | 1.13 | *−2.51* |
| 9320 | MYB-related protein 1 | MYB | 1.82 | 8.71 | **2.26** |
| 17730 | MYB-related protein 3r-1-like | MYB | 29.21 | 4.41 | *−2.73* |
| 16556 | MYB-related protein 3r-1-like | MYB | 32.22 | 5.36 | *−2.59* |
| 17731 | MYB-related protein 3r-1-like | MYB | 11.04 | 1.9 | *−2.54* |
| 40625 | MYBJ6 transcription factor | MYB | 20.44 | 3.47 | *−2.56* |
| 11797 | Telomere repeat binding factor 1 | MYB | 10.18 | 36.07 | **1.83** |
| 27346 | Two-component response regulator ARR2 | MYB | 0.3 | 4.73 | **3.98** |
| 23556 | SQUAMOSA promoter binding protein | SPL | 23.77 | 1.53 | *−3.96* |

**Table 3** (*continued*)

| Contig # | Gene name | TF class | Emergent RPKM | Mature RPKM | Log2FC (M/E) |
|---|---|---|---|---|---|
| 36093 | SQUAMOSA promoter binding protein | SPL | 31.84 | 4.88 | −2.71 |
| 10750 | SQUAMOSA promoter binding-like protein 12-like | SPL | 89.29 | 18.25 | −2.29 |
| 19993 | SQUAMOSA promoter binding-like protein 16-like | SPL | 14.65 | 2.5 | −2.55 |
| 36043 | SQUAMOSA promoter-binding-like protein 8 | SPL | 22.67 | 2.75 | −3.04 |
| 43513 | C2H2 zinc finger protein | ZF | 27.98 | 3.01 | −3.22 |
| 4333 | COL domain class transcription factor | ZF | 111.27 | 300.94 | **1.44** |
| 776 | CONSTANS protein | ZF | 188.15 | 483.34 | **1.36** |
| 27533 | ZF CCCH domain-containing protein | ZF | 9.97 | 1.37 | −2.86 |
| 53388 | ZF CCCH domain-containing protein 31 | ZF | 5.69 | 0.33 | −4.11 |
| 28812 | ZF CCCH domain-containing protein 66-like | ZF | 31.56 | 6.68 | −2.24 |
| 11427 | ZF protein nutcracker-like | ZF | 13.45 | 2.3 | −2.55 |
| 41587 | ZF-HD homeobox protein at4g24660-like | ZF | 22.94 | 4.01 | −2.52 |
| 25703 | ZF-HD homeobox protein at4g24660-like | ZF | 10.85 | 2.11 | −2.36 |
| 40970 | ZF-HD homeobox protein at4g24660-like | ZF | 79.33 | 17.74 | −2.16 |

underlying the shift from $C_3$ photosynthesis in young *Bienertia* leaves to the specialized $SCC_4$ photosynthetic mechanism in mature leaves (Fig. 2).

In emergent tissues, high expression of cell cycle regulators such as cyclins, cyclin-dependent kinases, and aurora kinases, along with cytoskeletal components like tubulins, kinesins, and microtubule-associated proteins, indicates the occurrence of active cell division and cytoplasmic reorganization. These processes are likely critical for establishing the cellular architecture required for chloroplast partitioning. Genes such as TOC and OEP80, involved in protein import into chloroplasts, suggest that foundational steps in chloroplast biogenesis, and perhaps differentiation, are initiated during this stage. Additionally, expression of BEL1-like regulates meristem identity, suggesting broader developmental signals coordinating shoot architecture and cellular differentiation. These findings suggest that early developmental processes active in emergent tissues lay the groundwork for the physical and functional separation of chloroplast types.

As leaves mature, the transcriptional profile shifts toward processes supporting $SCC_4$ functionality. Photosystem-related genes, including components of PSI, PSII, and light-harvesting complexes, are highly expressed, reflecting their roles in optimizing light capture and energy conversion. This is complemented by the expression of regulatory genes like SIG and RPO-D, essential for chloroplast-specific gene expression, as well as MgPME cyclase, which supports chlorophyll production. The upregulation of FeSOD indicates that oxidative stress management becomes particularly important in mature tissues, where high rates of photosynthesis generate reactive oxygen species.

Carbon fixation-associated enzymes also exhibit stage-specific expression patterns that shed light on the $C_3$–$SCC_4$ transition. The differential expression of PEPC transcripts in both emergent and mature tissues suggests functional specialization of this gene during development, with distinct isoforms potentially aligning with early metabolic priming and later bicarbonate fixation in peripheral chloroplasts. Similarly, elevated expression of CA

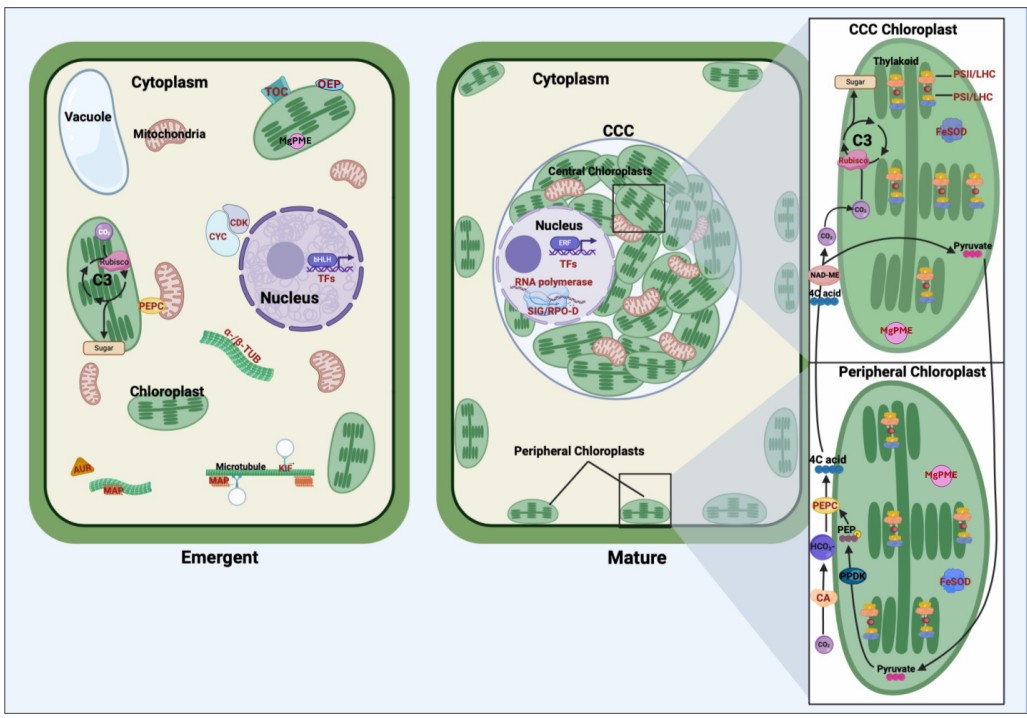

**Figure 2** **Comparative representation of young emerging and mature *Bienertia sinuspersici* leaf cells, highlighting organelles, carbon fixation pathways, and proteins corresponding to differentially expressed genes.** In young cells, genes associated with cell cycle progression and structural organization are highly expressed, including cyclins and cyclin-dependent kinases (CYC/CDK), alpha- and beta-tubulin (α-/β-TUB), microtubule-associated proteins (MAP), kinesin motor proteins (KIF), and aurora kinases (AUR). These genes facilitate cellular division, cytoskeletal organization, and early chloroplast biogenesis. Translocon of the outer chloroplast membrane (TOC) proteins and outer envelope proteins (OEP) support the import of nuclear-encoded proteins into chloroplasts, laying the groundwork for photosynthetic specialization. The general $C_3$ pathway is depicted in the young chloroplast. Transcription factors (TFs) regulate gene expression programs critical for cell proliferation, differentiation, and early leaf anatomical development. In mature cells, *Bienertia*'s specialized $SCC_4$ morphology is illustrated, featuring dimorphic chloroplasts within the distinct central cytoplasmic compartment (CCC) and at the cell periphery. Genes associated with photosynthesis and carbon fixation, including carbonic anhydrase (CA), phosphoenolpyruvate carboxylase (PEPC), and RUBISCO, display higher expression in mature cells. Proteins such as photosystem II (PSII), photosystem I (PSI), and light-harvesting complex (LHC) enhance light capture and energy conversion. At the same time, iron superoxide dismutase (FeSOD) mitigates oxidative stress generated by high photosynthetic activity. Sigma factors (SIG) and RNA polymerase D (RPO-D) regulate chloroplast-specific gene expression; magnesium-protoporphyrin IX monomethyl ester cyclohydrolase (MgPME cyclase) supports chlorophyll production; and various TFs, including ethylene-responsive TFs (ERFs), contribute to optimizing photosynthesis, stress response, and metabolic homeostasis. Arrows depict the flow of carbon intermediates within the $SCC_4$ pathway, highlighting the compartmentalized functionality of central and peripheral chloroplasts. Proteins corresponding to differentially expressed genes are labeled in red text.

and the RUBISCO small subunit in mature tissues reflects their roles in establishing the spatial and functional compartmentalization of $SCC_4$ photosynthesis, with peripheral and central chloroplasts specializing in initial and secondary $CO_2$ fixation, respectively.

Overall, this model highlights how the emergent stage establishes the cellular framework and initiates key molecular processes, while the mature stage refines and specializes these systems for efficient light harvesting and carbon fixation.

## CONCLUSION

The transcriptome analysis of emergent and mature leaf tissues in *Bienertia sinuspersici* has facilitated identification of possible candidate genes enabling the transition from $C_3$ to $SCC_4$ photosynthesis.

In emergent tissues, the enrichment of cell cycle regulators and cytoskeletal components highlights the prioritization of cellular proliferation, structural organization, and early chloroplast assembly, laying the foundations for intracellular compartmentalization and the establishment of central and peripheral chloroplasts. The upregulation of genes associated with photosynthesis and carbon fixation in mature tissues reflects the transition to a fully functional $SCC_4$ photosynthetic system. This sequential developmental progression illustrates how early structural and regulatory processes establish the cellular framework while mature tissues refine and specialize these systems to support $SCC_4$ functionality.

Although further time-course analyses and functional studies are necessary to elucidate the mechanisms driving the $C_3$–$SCC_4$ transition fully, this study provides a foundation for future research leveraging $SCC_4$ mechanisms to enhance crop productivity and resilience in the face of changing climates.

### Funding

This project was funded by a National Science Foundation Grant MCB 1146928 to Gerald E. Edwards and Amit Dhingra; WSU ARC Hatch project WNP0001, Texas A&M AgriLife Research Hatch Grant TEX0 9950, and startup funds from Texas A&M AgriLife Research and Texas A&M University to Amit Dhingra. The funders had no role in study design, data collection and analysis, decision to publish, or preparation of the manuscript.

### Grant Disclosures

The following grant information was disclosed by the authors:
National Science Foundation:  MCB 1146928.
WSU ARC Hatch project WNP0001.
Texas A&M AgriLife Research Hatch Grant: TEX0 9950.
Texas A&M AgriLife Research.
Texas A&M University.

### Competing Interests

The authors declare there are no competing interests.

## Author Contributions

- Richard M. Sharpe conceived and designed the experiments, analyzed the data, prepared figures and/or tables, authored or reviewed drafts of the article, and approved the final draft.
- Seanna Hewitt analyzed the data, prepared figures and/or tables, authored or reviewed drafts of the article, and approved the final draft.
- Gerald Edwards conceived and designed the experiments, authored or reviewed drafts of the article, and approved the final draft.
- Amit Dhingra conceived and designed the experiments, performed the experiments, analyzed the data, authored or reviewed drafts of the article, and approved the final draft.

## Data Availability

The datasets generated are available at NCBI BioProject: PRJNA340188, SRA SRP083068: SRX850921, SRX2056177, SRX2056178, SRX2056185, SRX2056186.

## Supplemental Information

Supplemental information for this article can be found online at http://dx.doi.org/10.7717/peerj.19282#supplemental-information.

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
