# Peer review of "Comparative transcriptome analysis of emerging young and mature leaves of Bienertia sinuspersici, a single-cell C4 plant"

_PeerJ, doi:10.7717/peerj.19282_

## Round 0.1 · original submission · Major Revisions

Both reviewers raised points that should be addressed. Additionally, each reviewer attached an annotated version to facilitate locating the points they assessed.

·

Basic reporting

The manuscript entitled "Comparative transcriptome analysis of emerging young and mature leaves of Bienertia sinuspersici, a single-cell C4 plant" aims to screen a global view of gene expression associated with the leaves' development, to provide some insights about the single-cell C4 plant type. The study is very interesting and could help elucidate some molecular players regulating C3-C4 photosynthesis, providing further biotechnology applications in crops.
The RNA-seq is a powerful approach providing a great quantity of expressed genes, as observed in the study. The authors found that most of the genes are related to cell division, cell cycle, and others regulating the development. However, there are some with great importance that could be more discussed in the results, for example, someone that I delighted in the attached pdf and can be found in Table 1, revealed by GO analysis (functions related to the anatomical structure development, thylakoid, etc).
I'd recommend also improving the results, which more graphs per example, a Heatmap to a global view of the distribution of gene expression between the two samples.
The most of comments I have inserted are in the attached PDF and listed below.

Experimental design

As the comments that I have inserted on the attached pdf, it would be better if some pictures were provided to illustrate the two studied samples (young and mature leaves), and the plant as well.
Also, it is need more information about the plants: how old are these plants? the samples collected in the juvenile or mature phase?

Validity of the findings

The SRA numbers from each library constructed were not provided, suggesting that the sequences were not deposited (yet) on the NCBI.

Additional comments

Line 30: It's a little bit confused... Did you mean emerging young and mature leaves? Like in the title?
Line 58: I guess that is a something typing mistake here
Line 69: Check if can group these two refs., like Voznesenskaya et al., 2002, 2003
Line 73: Change for the abb form (SCC4)
Line 73: Would be better if this paragraph, until this period, could be joined with the previous paragraph. And the last sentence merged with the next paragraph, then, sorted the topics.
Line 86: Specify NAD-ME
Line 87: could exemplify someones of these enzymes?
Line 94: there's lacking of the main question or hypothesis in this work. The author could explore more this issue in order to be more clear about the importance of this study related to the SCC4 plants. Also, after the aims, one or two sentences bring the main results in a general way and perspectives of these findings.
Line 96: The Intro section needs to have more recent articles about this theme or species.
Line 104: from the same plant? How old are these plants?
Line 106: Could be provide some pictures of these two types of leaves?
Line 121: until the 75% DEPC-ethanol is evaporated? My question is because, in the next sentence, the pellets were suspended in RNAse-free water....
Line 124: DNAse, Insert the manufacturer
Line 155: using the FASTQC tool? Which tools were used to trim and/or remove the adaptors?
Line 165: It's important to mention that there's no available genome for this species (if is true) and, if this is the case, a reference transcriptome or genome from a related/close species.
Line 222: Since Differentially Expressed Genes are not discussed in this section (but in the next section), they should removed from the header
Line 251-256: Insert some reference to support these two statements
Line 259: could be discussed with the genes identified in this paper, some comparing with the present study
Line 261-263: Needs some reference
Line 269: this term it's a quite bit repeated
Line 286-294: Although this phenomenon is evident and could be associated with the dimorphic chloroplasts of Bienertia, this paragraph doesn't show some discussion with results from the transcriptome realized. Per example, did the authors find genes related to this cell process? If yes, they could show the genes here.
Line 339-341: how about genes related to C3-C4 or SCC4 modules?
Line 355-359: Needs reference here
Line 369: It could merge with the previous sentence, per example: "... SCC4 photosynthesis in Bienertia (Han et al. 2023), which performed transcriptome ..."
Line 378: Check the request format by the PeerJ in this citation.
Line 382: What are the main genes? Anyone from these have been identified in this study?

Reviewer 2 ·

Basic reporting

Clear and unambiguous, professional English used throughout. There could be more mention and contextualization in terms of literature cited, both in terms of B. sinuspersici literature and also in terms of other species with SCC4. Table are professional, no figures are presented, but I suggest to include some in the revised version. The result and hypothesis are self-contained in the scope of a transcriptomic work, but anatomy, biochemistry and morphology are left for the reader to find in other articles. More in the reviewer comments.

Experimental design

Original primary research within Aims and Scope of the journal. Research question well defined, relevant & meaningful. It could be improved how this research fills an identified knowledge gap by comparing to previous work in the group and by explaining how SCC4 contributes to C3 engineering more clearly. More in the reviewer comments.

Validity of the findings

Novel and impactful findings but more on CCM-related genes could be addressed. Replication described as pooled sample and the use of software modelling to create technical replication - reviewer suggest to better explain this decision. Data provided is of quality and sound, although the use of technical replication is slightly worrying for reproducibility. Conclusions are well stated, linked to original research question & limited to supporting results

Additional comments

Various major and minor comments are included in the attached pdf. Please see the file for explanations.

Annotated reviews are not available for download in order to protect the identity of reviewers who chose to remain anonymous.

---

## Round 0.2 · Minor Revisions

Both referees raised minor issues, although they also agree that the manuscript has largely improved in this version. I do agree with all observations, that should be addressed in a next version.

·

Basic reporting

The authors have strongly improved the manuscript and the text is now better and clearer. I have only minor considerations about the manuscript, which are listed below.

line 222-223 - The CLC Trim Sequence process was used to trim the 454
223 read datasets for a Phred value of 15

lines 275-279 - The sentences are duplicated.

Figure 1 - Just an aesthetic point, add the lines corresponding to the axes for a better view.

Table 2 - The column Log2FC, describes E/M, but in the legend is the opposite. Please, indicate which is the correct one. In addition, in the sentence "and those corresponding to genes that were significantly upregulated in the mature tissues (value < 1) are in bold.", did you mean value > 1? In this case, the up-regulated genes in mature tissues are positive values.

line 373 and Table 2 - this is an interesting result. But in table 2, there are two ERECTA citations. Is one corresponding to the ERL? If yes, indicate it.

line 385 - change WAKL9 to italic form.

Experimental design

no comment

Validity of the findings

no comment

Reviewer 2 ·

Basic reporting

a- Clear and unambiguous, professional English used throughout. b- Literature references, sufficient field background/context provided. c- Professional article structure, figures, tables. Raw data shared. d- Figure 2 was a great addition to the work. Figure 1 is a bit confusing. Any reason for presenting log10FC? How about replacing this (which can just be in the text as it already is and is clear enough) with a venn diagram showing the number of genes exclusive to each leaf age and the number shared? It doesn't necessarily need the information about DEGs at this point. e- Overall goal can be adjusted as suggested in further comments.

Experimental design

a- Original primary research within Aims and Scope of the journal. b- Research question can be refined, but is relevant & meaningful. See further comments. c- It is stated how research fills an identified knowledge gap, but it needs adjusting. d- Rigorous investigation performed to a high technical & ethical standard. e- Methods described with sufficient detail & information to replicate.

Validity of the findings

a- Impact and novelty are assessed. Replication issue should be presented further, see further comments. Benefit to literature is clearly stated. c- All underlying data have been provided; they are robust, statistically sound, & controlled. d- Conclusions are well stated, linked to original research question & limited to supporting results. There are points for adjustment though, see further comments.

Additional comments

Thank you for submitting the reviser work. The points raised by the reviewers were clearly addressed and the improvement is noticeable. I bring a few more point for consideration:

Abstract lines 20-21 I don’t think authors need to write C3-type and C4-type, just C3 and C4 should be enough.

Line 80 correct subscript

Line 143 no need to repeat the citation, just include (2023) in parenthesis.

Line 244-255 I think it would be insightful for the reader to understand what was the problem with having sufficient material for real biological replicates. Is the plant difficult to sample in the wild and then also difficult to cultivate in greenhouse/climate chamber? I believe it would be helpful to understand why wasn’t there more material available.

Line 250 “to compare the treatments at each time point”. I am confused, which treatments at which timepoints? I understand two tissues sampled 2h after lights are on.

Line 553 The transcriptomic work presented is definitely novel, insightful, and nicely discussed, but I am not sure I would use the word “comprehensive” here, due to the limited biological replication and also the need to include intermediate stages between full C3 and full SCC4 states. How about: “we present an in-depth discussion on possible candidate genes enabling the development from C3 to SCC4”? Something is this line would be more accurate. The tone adopted in the results+discussion is super appropriate though and the text reads much better now.

Line 565 subscript

Lines 142-154 After rereading, I came back to the page stating the goals and I would still suggest further rephrasing. Yes, it would be essential to unravel the “precise molecular factors driving the establishment of dimorphic chloroplasts and the transition from C3 to SCC4 photosynthesis”, as the authors state. But that was not addressed here. It would be addressed in a more detailed time course really showing the intermediate stage in forming that cytoplasmic compartment. I would argue this is definitely a knowledge gap and the current work addresses this issue, but not exhaustively and comprehensively. It can be the first time this issue is addressed and your findings can surely serve as basis for a future work, to keep track of these cytoskeleton and cell cycle genes and see how they coordinate the change to SCC4. However, here more static results are found: in emergent leaves it’s like this, in developed it’s like that, without a transition. I would kindly ask you to smoothen the phrasing in this part. Particularly in line 153, I would place it in the future, “these finding will have a broad impact”, since improving crop productivity was not addressed here.

---

## Round 0.3 · Minor Revisions

Reviewer 1 still raises some minor points thet weren't uptaded.

·

Basic reporting

The authors have addressed almost all issues pointed out by reviewers. However, the issues below, from the last review, didn't change in the files, although the authors mentioned an update. Check if occurred a mistake at the moment of uploading the files. Since these are fixed, I recommend the publication of this manuscript.

The previous issues that weren't updated:
3. Figure 1 - Just an aesthetic point, add the lines corresponding to the axes for a better view.
Response: We have added the axis lines as suggested (see Figure 1)
4. Table 2 - The column Log2FC, describes E/M, but in the legend is the opposite. Please, indicate which is the correct one. In addition, in the sentence "and those corresponding to genes that were significantly upregulated in the mature tissues (value < 1) are in bold.", did you mean value > 1? In this case, the up-regulated genes in mature tissues are positive values.
Response: Thank you for identifying this issue. We have adjusted the legend to match the table (see Table 2 legend).
5. Line 373 and Table 2 - this is an interesting result. But in table 2, there are two ERECTA citations. Is one corresponding to the ERL? If yes, indicate it. 
Response: Yes, contig 3912 is an ERL gene. We have updated the Table 2 accordingly.

Experimental design

no comments

Validity of the findings

no comments

Reviewer 2 ·

Basic reporting

Everything addressed in previous revision rounds.

Experimental design

Everything addressed in previous revision rounds.

Validity of the findings

Everything addressed in previous revision rounds.

Additional comments

Abstract line 42 - remove "-type" from: C3-type crops into C4 type

---

## Round 0.4 · accepted · Accept

I have assessed the revised version of this manuscript along with the authors’ detailed response to the reviewers’ comments. I confirm that the authors have adequately addressed all points raised during the peer-review process. The manuscript is improved and now meets the publication criteria. I recommend its acceptance for publication in the present form.